# Economic disparity among generations under the Paris Agreement

Haozhe Yang [1] & Sangwon Suh [1✉]

The costs and benefits of climate change mitigation are known to be distributed unevenly across time and space, while their intergenerational distribution across nations has not been evaluated. Here, we analyze the lifetime costs and benefits of climate change mitigation by age cohorts across countries under the Paris Agreement. Our results show that the age cohorts born prior to 1960 generally experience a net reduction in lifetime gross domestic product per capita. Age cohorts born after 1990 will gain net benefits from climate change mitigation in most lower income countries. However, no age cohorts enjoy net benefits regardless of the birth year in many higher income countries. Furthermore, the cost-benefit disparity among old and young age cohorts is expected to widen over time. Particularly, lower income countries are expected to have much larger cost-benefit disparity between the young and the old. Our findings highlight the challenges in building consensus for equitable climate policy among nations and generations.

[1] Bren School of Environmental Science and Management, University of California, Santa Barbara, CA, USA. ✉email: suh@bren.ucsb.edu

Younger generations emerged at the forefront of the global climate movement in recent years[1,2]. One of the prevailing narratives to this phenomenon is that younger and future generations are the greatest victims of climate change driven by the actions and inactions of older generations[3–5]. Supported by such narratives, some studies indicated the presence of inter-generational gaps in the perceptions toward climate change mitigation[2,6–8]. Several studies have explored how the economic policy can be designed to reduce or eradicate the intergenerational disparity in climate change mitigation[9,10].

Though many studies have discussed the justice and inequality issues among generations under climate change[3,11,12], there were, however, no peer-reviewed literature that quantifies the costs and benefits of climate change mitigation by age cohorts at a country level.

In this study, we quantify the lifetime costs and benefits of climate change mitigation by age cohorts across countries under the Paris Agreement. In this paper, the cost of climate change mitigation refers to the gross domestic product (GDP) loss compared to the counterfactual scenario without climate change mitigation[13]. To measure the loss of GDP, Integrated Assessment Models (IAMs) are developed by many research groups to couple energy, economy, and climate (Supplementary Note). In these IAMs, the economic modules generally follow general or partial equilibrium models[14]. Here, the data for the cost of climate change mitigation is derived from several IAMs[15] in the 2014 IPCC report. According to the report, the abatement cost of climate change mitigation range 2–6% of global GDP by 2100 relative to pre-Paris Agreement policy[13].

The benefit of climate change mitigation refers to the avoided economic damage by stabilizing global temperature[16]. Burke et al.[17] developed a damage function that measures the nonlinear relationship between temperature and economic growth (BHM damage function). Using this nonlinear relationship, Burke et al. estimated that keeping the global temperature at 2010 level could save 23% of global GDP by 2100[17]. Though the Burke method is still under discussion[18,19], this empirical nonlinear GDP–temperature relation has been widely applied in the cost-benefit analysis of climate change mitigation[16,20,21].

The climate change mitigation scenarios under the Paris Agreement employed in our models do not consider the policies to address the intergenerational disparity. The costs and benefits of climate change mitigation are modeled for the period of 2020–2100 (Supplementary Fig. 1). The benefit of climate change mitigation hereafter is quantified by the BHM damage function, and the cost of climate change mitigation is calculated by assuming a triangle distribution of GDP loss reported by the 2014 IPCC report. To quantify the cost-benefit disparity, we estimate the lifetime costs and benefits at a 3% discount rate by age cohort in 169 countries under the 2 °C target of the Paris Agreement. The lifetime cost and benefits are measured as accumulative GDP per capita (in 2018 dollars) during the lifetime of an age cohort. The lifetime of an age cohort is calculated by using the expected life expectancy for the age group[22]. The distribution of GDP per capita across age cohorts follows the income distribution from the OECD database. The Paris Agreement scenario is represented by the Representative Concentration Pathways (RCP) 2.6. The Pre-Paris Agreement scenario, which is the baseline scenario in our study, is represented by the Shared Socioeconomic Path (SSP) 4 and RCP 6.0[23] in the main text (analysis of other SSP and RCP scenarios can be found in "Data Availability").

## Results

**Costs and benefits over generations.** We first evaluate the change of lifetime GDP per capita for age cohorts born between 1920 and 2020. Our results show that climate change mitigation incurs a net reduction in lifetime GDP per capita for age cohorts born prior to 1960 across nearly all nations (Fig.1). In low-income countries, the age cohorts born before 1960 incur the largest reduction of average lifetime GDP per capita compared to the same age cohorts in countries with higher income. In low-income countries, the age cohort born between 1920 and 1960 is estimated to incur, on average, ~2.5% net reduction in lifetime GDP per capita under the Paris Agreement (Fig. 1d, h). In contrast, in high-income countries, the same age cohorts incur the least net reduction (<1%) in average lifetime GDP per capita.

In most of the lower-middle-income and low-income countries, age cohorts born after 1990 will start to have a net gain of lifetime GDP per capita in the course of climate change mitigation under the Paris Agreement. By quantity, the net gain of lifetime GDP per capita among the younger age cohorts is asymmetrically larger than the net reduction among the older age cohorts. In low-income countries (Fig. 1d, h), the age cohort born in 2020 enjoys a net gain of ~6% in lifetime GDP per capita on average, while the age cohort born in 1950 incurs a net reduction of ~3%. In lower-middle-income countries (Fig. 1c, g), on average, the largest net gain of lifetime GDP per capita is 5–8-folds larger than the net reduction in absolute value.

In high- and upper-middle-income countries, the trend of lifetime GDP per capita by age cohort is sensitive to the model specifications that measure the benefits of climate change mitigation. When using the short-term BHM damage function to measure the lifetime benefits (short-term benefits), which is commonly used in other research[16,17,20,24,25], the age cohorts in many high- and upper-middle-income countries still incur a net reduction of lifetime GDP per capita (short-term net benefits) with the progression of birth year, including the age cohorts born in 2020. The age cohorts in high and upper-middle-income countries, on average, barely gain any net benefits throughout the birth years considered (Fig. 1a, b). On average, all age cohorts in high-income countries lose 0–2% of lifetime GDP per capita, and those in upper-middle-income countries lose 0–3% of lifetime GDP per capita.

However, when using the long-term BHM damage function to measure the lifetime benefits (long-term benefits), the net gain of GDP per capita (long-term net benefits) increases in high- and upper-middle-income countries with the progression of birth years. The age cohorts born in 1960 in high-income countries (Fig.1e), and the age cohorts born in 1980 in upper-middle-income countries (Fig. 1f) start to show net gains of average lifetime GDP per capita under climate change mitigation.

The uncertainty range of the long-term BHM damage function is much wider than that of the short-term damage function (Supplementary Fig. 2). Due to the large uncertainties of the long-term damage function and the lack of robust evidence for the long-term benefits, the short-term benefits of climate change mitigation are more commonly discussed in the current literature[19,24,25].

**Breakeven generation.** We define a breakeven generation as the age cohort that breaks even the lifetime cost and benefit from climate change mitigation under the Paris Agreement in a given year studied. An age cohort born after the breakeven generation will gain net benefits from climate change mitigation, and an age cohort born before this breakeven generation will bear net costs. In 2020, more than three-quarters of the population are born after the breakeven generation in Latin America, South Asia, and Western Asia (Fig. 2e and Supplementary Fig. 3e). In Latin America and South Asia, the breakeven generations are born prior to 1970 (Fig. 2 and Supplementary Fig. 3), which are the

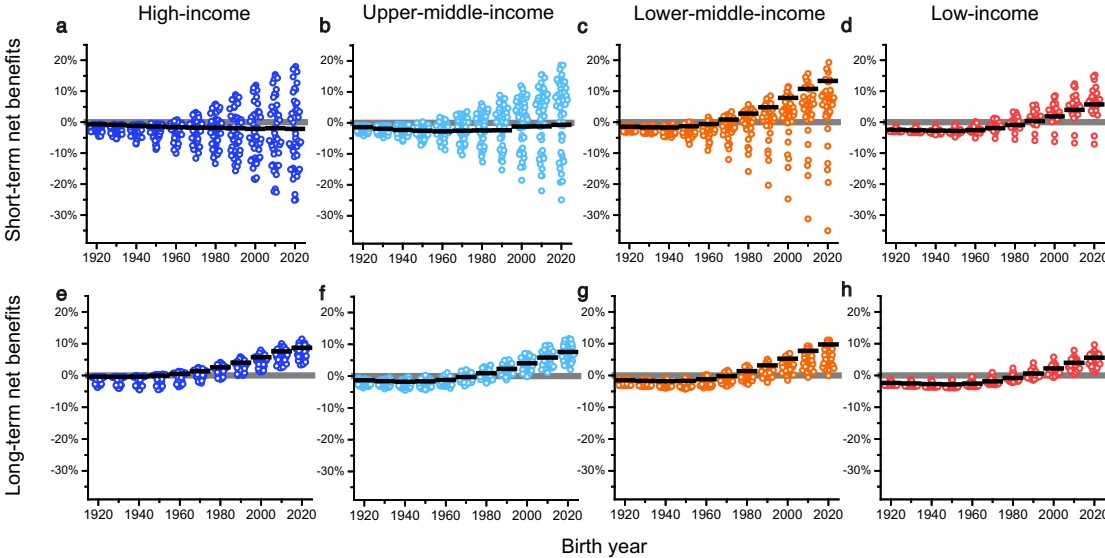

**Fig. 1 Percentage change of lifetime GDP (gross domestic product) per capita for age cohorts born during 1920–2020.** Using the short-term BHM (Burke, Hsiang, and Miguel) damage function to calculate the lifetime benefits of climate change mitigation (short-term benefits), the percentage change of lifetime GDP per capita (short-term net benefits) for age cohorts born during 1920–2020 is shown in **a** high-income countries, **b** upper-middle-income countries, **c** lower-middle-income countries, and **d** low-income countries. Using the long-term BHM damage function to calculate the lifetime benefits of climate change mitigation (long-term benefits), the percentage change of lifetime GDP per capita (long-term net benefits) for age cohorts born during 1920–2020 is shown in (**e**) high-income countries, **f** upper-middle-income countries, **g** lower-middle-income countries and **h** low-income countries. The solid black lines represent the population-weighted average of the percentage change of lifetime GDP per capita for each age cohort. For age cohorts with the same birth year, the average percentage change of lifetime GDP per capita is quantified in $n = 47$ high-income countries, $n = 50$ upper-middle-income countries, $n = 43$ lower-middle-income countries, and $n = 29$ low-income countries. A circle symbol represents an age cohort in a country. The color of circles represents the income group of a country. See details in Supplementary Data.

earliest across the world. However, in Eastern Europe, only the age cohorts born after 1980 enjoy a net benefit from climate change mitigation, so that more than half of the current population in that region are born before the breakeven generation (Fig. 2e and Supplementary Fig. 3e).

In high-income countries, the breakeven generations are born prior to 1980 in Spain, Australia, and Saudi Arabia (Fig. 2a). The birth years of the breakeven generation in Europe, Canada, and the United States are sensitive to the model specification. Using the short-term net benefits, none of the age cohorts studied (age cohorts born between 1920 and 2020) breaks even the costs and benefits of climate change mitigation in Canada and most Western European countries, and the breakeven generation in the United States is born in 1994. This is because, in colder regions, the temperature increase has neutral or positive effects on the economy in the short-term[24,26–28]. When considering the long-term net benefits, the breakeven generation are born prior to 1970 in Canada, the United States, and Western Europe, and more than three-quarters of the current population are born after the breakeven birth year (Supplementary Fig. 3).

Among the upper-middle-income countries (Fig. 2b), the breakeven generations are the youngest (born after 1990 or nonexistent) in Russia and South Africa, while they are the oldest in Latin American countries (born before 1970). In Russia, only 0–31% of the current population are born after the breakeven generation (Fig. 2e and Supplementary Fig. 3e). In contrast, in Brazil and Mexico, >75% of the current population are born after the breakeven generation. The birth years of the breakeven generation in Asia are uncertain due to the model specification, ranging from 1970 to infinity (nonexistence). In China, none of the age cohorts breakeven the costs and benefits when using the short-term net benefits, but two-thirds of the current population are born after the breakeven birth year when using long-term net benefits.

In lower-middle- and low-income countries (Fig. 2c, d), the breakeven generation are generally born before 1990. As the population is younger in lower-middle and low-income countries, more than half of the current populations are born after the breakeven birth year in the majority of these countries (Fig. 3e and Supplementary Fig. 3e). The breakeven generations are the oldest in South Asia and Latin America; >75% of the population are born after the breakeven generation. In India, Pakistan, and Bolivia, the breakeven generation are all born between 1950 and 1970. In Southeastern Asia, the breakeven generations are born before 1980. The breakeven generations in Africa are born during 1980–1990, and are 10–30 years younger than Latin America and South Asia.

**Future intergenerational cost-benefit disparity.** The intergenerational disparity index (IDI) is calculated in this paper as the percentage change in lifetime GDP per capita among the 25-year-old age cohort minus that among the 75-years-old age cohort. For example, if the 25-year-old age cohort gains 10% of lifetime GDP per capita, and the 75-year-old age cohort loses 10% of lifetime GDP per capita, the intergenerational disparity index is 0.2 (0.1 − (−0.1)). A larger absolute value of IDI in a country indicates that the economic disparity between the young and the old under climate change mitigation is severe.

Here, we project IDIs from 2020 to 2100, and find that the intergenerational disparity is widening using the short-term net benefits. Specifically, IDIs become larger in countries with lower income over time (Fig. 3). In 2020, IDIs are <0.1 in most countries (Fig. 3b). After 2020, the intergenerational disparity grows significantly, particularly in countries with lower income. In 2020, the median IDI is 0.05 for lower-middle-income countries and 0.06 for low-income countries. However, in 2100, the median of IDI increases to 0.42 for lower-middle-income countries and 0.52 for low-income countries. For upper-middle-

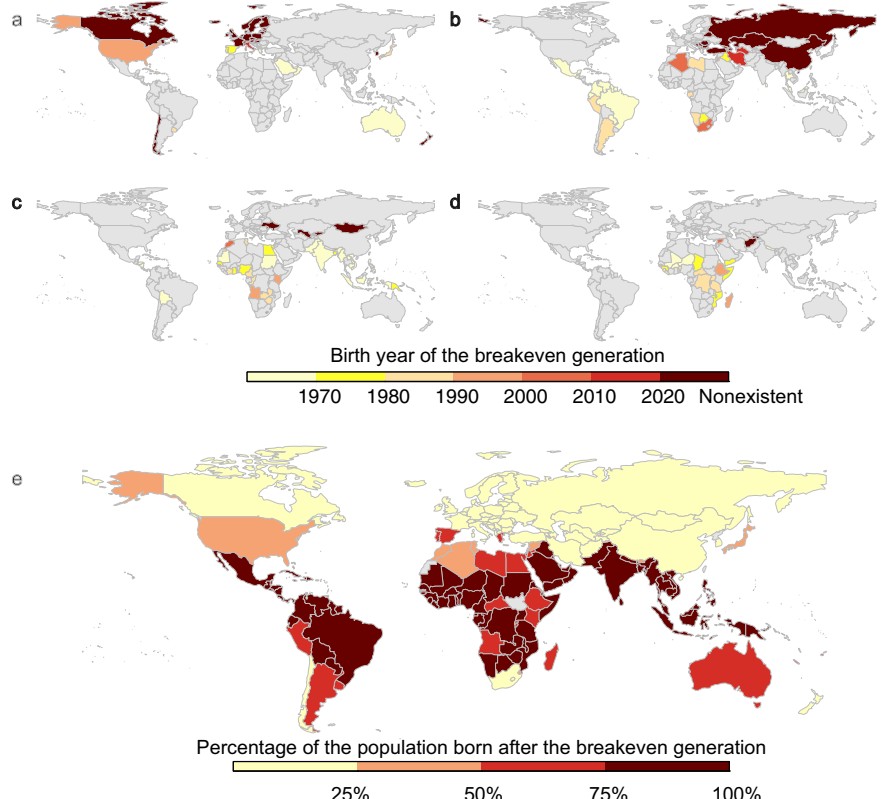

**Fig. 2 Breakeven generation and the percentage of the population born after the breakeven generation using the short-term net benefits.** The birth year of the breakeven generation in 2020 in **a** high-income countries, **b** upper-middle-income countries, **c** lower-middle-income countries, and **d** low-income countries. **e** The percentage of the population born after the breakeven generation. In (**a**–**d**), different colors represent different ranges for the birth years. In (**e**), different colors represent different ranges for the percentage of the population. Here, we use the short-term benefits to measure the lifetime benefits. We use the long-term benefits in Supplementary Fig. 3. See details in Supplementary Data.

income countries, IDI also increases; the median value increases from 0.05 in 2020 to 0.27 in 2100.

For high-income countries, the intergenerational disparity is the smallest compared with other income groups. From 2020 to 2100, IDI remains within −0.25 to 0.25 for most high-income countries. In 2100, the median IDI for high-income countries is only −0.04. In many high-income countries, the negative IDI indicates that the climate change mitigation under the Paris Agreement favors older age cohorts than younger age cohort in terms of lifetime GDP per capita.

On a global scale, the intergenerational disparity is widening the most in Latin America, Africa, and Western and Southern Asia (Supplementary Fig. 4). From 2020 to 2100, IDIs in these countries increase from less than 0.1 to over 0.25. Furthermore, in Saudi Arabia, Sudan, Niger, and Mauritius, IDI is over 1, whereas they remain within less than 0.25 in Eastern Asia, Europe, and North America.

The use of long-term net benefits results in a similar trend (Supplementary Fig. 5). IDIs are increasing over time, and they become larger in countries with lower income. In low-income countries, the median IDI increases from 0.04 (2020) to 0.34 (2100). In lower-middle-income countries, upper-middle-income countries, and low-income countries, the median IDI increases from 0.05 to 0.31, 0.28, and 0.23 during 2020–2100.

### Discussion

In this paper, we find a large cost-benefit disparity among age cohorts under the Paris Agreement. On a global level, the older age cohorts born before 1960 hardly gain in climate change mitigation, while younger age cohorts born after 1990 are gaining large net benefits. This result indicates that younger generations

may be more strongly motivated to mitigate climate change, which is well-aligned with the prevailing narrative that tries to explain the rise of the younger generation in the global climate movement.

However, country-level analysis paints a somewhat more complex picture. Our results based on the short-term damage function of climate change, for example, show that no age cohorts enjoy net benefit from climate change mitigation in most Western European countries in 2020. Therefore, the rise of the younger generation among Western European countries in climate movement cannot be explained by the economic self-interest under the short-term damage function, while using the long-term damage function, we find that younger age cohorts in Western Europe also benefit from climate change mitigation.

In addition, our results may provide an insight on the attitude toward climate change mitigation. Should the level of support to climate change mitigation be positively correlated to the net lifetime benefits from climate change mitigation, lower-income (lower-middle- and low-income) countries are likely to see more support to climate change mitigation from older generations, because more than half of the current population are born after the breakeven generation in most low-income countries. Likewise, the climate change mitigation effort is likely to face challenges in Eastern Europe, because, regardless of the model specification, less than half of the current population in Eastern Europe are likely to gain net benefits from climate change mitigation.

Furthermore, our results also show that the cost-benefit disparity between the old and the young under climate change mitigation is widening in almost all countries over time. Although all age cohorts may gain from climate change mitigation, the benefits of younger age cohorts are much larger than that of older age cohorts. By 2100,

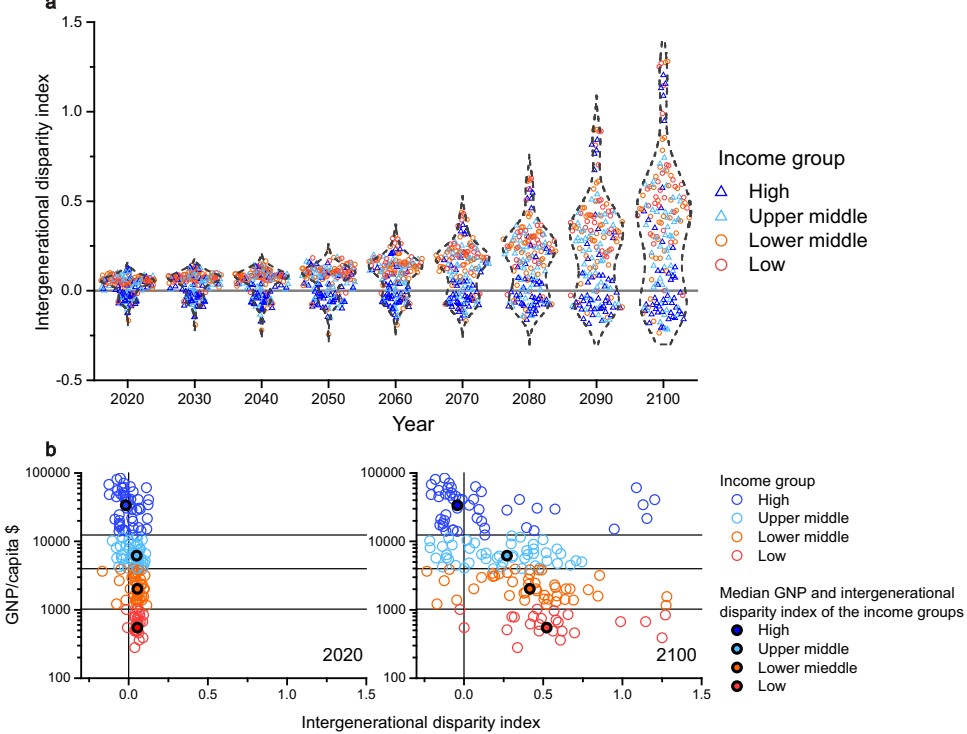

**Fig. 3 Intergenerational disparity among high-, upper-middle-, lower-middle-, and low-income countries using the short-term benefits. a** The distribution of the intergenerational disparity index from 2020 to 2100. **b** Intergenerational disparity index and GNP (gross national product) per capita (using 2018 GNP per capita) in 2020 and 2100. In (**a**), a triangle symbol represents a higher-income country, and a circle symbol represents a lower-income country. In (**b**), a circle symbol represents a country, and a solid circle with the black edge represents the median GNP and the median intergenerational disparity index of an income group. In (**a**, **b**), the color of a symbol represents the income group of a country. The intergenerational disparity index is calculated as the percentage change in lifetime GDP per capita among the 25-year-old cohort minus that among the 75-year-old age cohort. Here, the intergenerational disparity index is calculated using the short-term benefits. See details in Supplementary Data.

the intergenerational disparity in most countries is over fivefold larger than that in 2020. Particularly, we find that countries with lower-income experience larger intergenerational disparity over time. The widening intergenerational disparity in the costs and benefits of climate change mitigation indicates that building consensus across generations on climate policy may not become any easier in the future. Our results provide a new insight on intergenerational equity of climate change mitigation. Closing the economic disparity among age cohorts may require different climate policies to different age cohorts. The increase in renewable asset price may alleviate the intergenerational disparity under climate change mitigation, given that different age cohorts hold varying amounts of renewable assets[29]. Our study shows that the cost-benefit distribution among age cohorts can be an important consideration for policy makers when designing tax and fiscal policies in response to climate change mitigation.

## Methods

**Lifetime of age cohorts.** The lifetime of each cohort is calculated as the life expectancy after 2020. The age-specific life expectancy is derived from United Nations, and the country-specific life expectancy at birth is derived from World Bank. The following equations are used to derive the age- and country-specific lifetime after 2020.

$$SL = \begin{cases} SY - A, & \text{if } SL \geq 2020 \\ 2020, & \text{if } SL < 2020 \end{cases} \quad (1)$$

$$EL = \begin{cases} SY + E, & \text{if } EL \leq 2100 \\ 2100, & \text{if } EL > 2100 \end{cases} \quad (2)$$

SL is the start year of the lifetime for a studied age cohort. SY is the year studied, which is chosen from 2020 to 2100. $A$ is the age of the age cohort. EL is the end

year of the lifetime for a studied age cohort. $E$ is the life expectancy of the studied age cohort. The life expectancy is collected from World Bank[30] and World Population Prospects[22].

**Income distribution.** The income distribution across age cohort depends on the disposable income at each age group and the mean disposable income of the total population.

$$I_{t-T,t,i} = \frac{D_{t-T,i}}{\bar{D}_i} \quad (3)$$

where $t$ is the future year. $T$ is the birth year of the age cohort. $t$-$T$ is the age of an age cohort in year $t$. At year $t$, $D_{t-T,i}$ is the income of an age cohort born in year $T$ in country $i$. $\bar{D}_i$ is the average income in country $i$. At year $t$, $I_{t-T,t,i}$ is the ratio of the income for an age cohort born in year $T$ versus the average income of the total population in country $i$. The data for the income distribution across age cohorts are collected from the OECD database[31]. The income distribution of some developing countries is not included in the OECD database. Their income distribution is assumed to be the median of the income in developing countries that are included in the OECD database (Supplementary Table 1). We assume the income distribution is consistent over time.

**Calculation of benefits of climate change mitigation.** We followed the procedure in Burke et al.[17] (BHM damage function) to estimate the social benefits of climate change mitigation from 2020 to 2100, and the details are described below.

Global warming is measured by the global temperature increase between the pre-industrial level and 2100 period. The climate models in phase five of the Coupled Model Intercomparison Project (CMIP5)[32] generate the baseline temperatures from the mid-1800s to 2005 using historical radiative forcing and the future temperatures in the twenty-first century using radiative forcing under the RCP2.6, RCP4.5, RCP6.0, and RCP8.5 scenarios. Multiple global climate models are used to simulate each RCP scenario, and the average temperature increase in the models is used to represent the temperature increase in each of the RCP scenarios. Following the IPCC protocols, we use the years 1986–2005 as the baseline period and 2081–2100 as the RCP future period. According to the IPCC report[33], the temperature increase between the pre-industrial (1850–1900) level

and the current period (2003–2012) is 0.8 °C. Therefore, the projected global warming in 2100 relative to the pre-industrial level is calculated as

$$\Delta T_r = \frac{\sum_{m=1}^{N_r} \Delta T_{m,r}}{N_r} \tag{4}$$

$$\Delta T_{\text{pre}} = \Delta T_r + 0.8 \tag{5}$$

$\Delta T_{m,r}$ is the temperature increase between the 1986–2005 period and the 2081–2100 period under RCP scenario $r$ using model global climate model $m$, $N_r$ is the number of models used to simulate RCP scenario $r$, $\Delta T_r$ is the average temperature increase between the 1986–2005 period and the 2081–2100 period under RCP scenario $r$, and $\Delta T_{\text{pre}}$ is the global warming relative to the pre-industrial level. We chose the RCP2.6 scenario as the mitigation scenario because the average global temperature increase in all the models for this scenario is most consistent with the target to limit the global temperature increase to 2 °C.

Following the methods in Burke et al.[17] and Ricke et al.[20], we corrected the projected temperature in all RCP scenarios using the following correction equation:

$$\text{Temp}_{i,t,r} = \overline{\text{Temp}_i} + \frac{t - 2010}{2100 - 2010} \times \Delta(T_{i,r}) \tag{6}$$

where $\text{Temp}_{i,t,r}$ is the corrected temperature for country $i$ in year $t$ under RCP scenario $r$, $\overline{\text{Temp}_i}$ is the average temperature of the observations in country $i$ from 1980 to 2010, and $\Delta(T_{i,r})$ is the projected temperature increase between the 1986–2005 period and 2081–2100 period in country $i$ and RCP scenario $r$.

The effect of temperature on the GDP growth rate (BHM damage function) is described as

$$h(\text{Temp}_{i,t,r}) = \beta_1 \text{Temp}_{i,t,r} + \beta_2 \text{Temp}_{i,t,r}^2 \tag{7}$$

where $h(\text{Temp}_{i,t,r})$ is the effect of temperature on the GDP growth rate in country $i$ at year $t$ under RCP scenario $r$. The parameters $\beta_1$ and $\beta_2$ were estimated by Burke et al.[17] using historical country-level temperature and economic data.

The additional effect of warming on growth in year $t$ is calculated as

$$\delta_{i,t} = h(\text{Temp}_{i,t}) - h(\overline{\text{Temp}_i}) \tag{8}$$

where $\delta_{i,t}$ is the predicted additional effect of warming on GDP growth in country $i$ in year $t$.

The social benefits under climate change mitigation are calculated as:

$$G_{i,t,s,r} = G_{i,t-1,s,r}(1 + \eta_{i,t,s} + \delta_{i,t,r}) \tag{9}$$

$$B_{a,i,t,s,r} = (G_{i,t,s,r_{\text{mitigation}}} - G_{i,t,s,r})I_{a,t} \tag{10}$$

$$\text{CB}_{i,T,s,r} = \sum_{t=\text{SL}}^{\text{EL}} \left[ B_{i,t,s,r} \cdot I_{t-T,t,i} \times \prod_{\text{SL}}^{t} \frac{1}{1 + \text{Dis}_{i,t,s,r}} \right] \tag{11}$$

$$\text{Dis}_{i,t,s,r} = \rho + \mu G_{i,t,s,r} \tag{12}$$

$G_{i,t,s,r}$ is the GDP per capita in country $i$ in year $t$ (2020–2100) under SSP scenario $s$ and RCP scenario $r$, $\eta_{i,t,s}$ is the growth rate of GDP per capita under SSP scenario $s$. $B_{i,t,s,r}$ is the annual social benefit of country $i$ in year $t$ under SSP scenario $s$ and RCP scenario $r$. $G_{i,t,s,r_{\text{mitigation}}}$ is the GDP per capita in country $i$ in year $t$ under the mitigation scenario, and the mitigation scenario used in our research is RCP2.6 and SSP4. $\text{CB}_{i,T,s,r}$ is the discounted lifetime benefits of the age cohort born in year $T$ from country $i$ by achieving the mitigation scenario from the SSP scenario $s$ and RCP scenario $r$. $\text{Dis}_{i,t,s}$ is the discount rate in country $i$ in year $t$. Based on the classification of the World Bank[34], we classified the countries into four income groups: a high-income group, upper-middle-income group, lower-middle-income group and low-income group.

As shown in Eq. (16), the discount rate is determined by the Ramsey endogenous rule[35], where $\rho$ is the pure time preference and $\mu$ is the elasticity of marginal utility. If $\mu = 0$, Eq. (11) estimates the social benefits with a fixed discount rate. If $\mu \neq 0$, Eq. (11) estimates the social benefits with a growth-adjusted discount rate.

The growth rate of GDP ($\eta_{i,t,s}$) is derived from the SSP database developed by the International Institute for Applied System Analysis[36]. We used the code and compiled datasets from Burke Lab (https://github.com/burke-lab/BDD2018) to derive the population-weighted temperature increase from 2010 to 2100 at the country level. The country-level baseline temperatures in 2010 and the original code for calculating the benefits of climate change mitigation under multiple SSP and RCP scenarios are compiled from https://country-level-scc.github.io/.

**Calculating the cost of climate change mitigation.** We used the loss of GDP from the 2014 IPCC report to calculate the cost of climate change mitigation.

$$C_{i,t,s,r} = G_{i,t,s,r} \cdot L_{i,t,r_{\text{mitigation}}} \tag{13}$$

$$L_{i,t,r_{\text{mitigation}}} = L_{t,r_{\text{mitigation}}} \cdot R_{k,r_{\text{mitigation}}}, i \in k \tag{14}$$

$$\text{CC}_{i,T,s,r} = \sum_{t=\text{SL}}^{\text{EL}} \left[ C_{i,t,s,r} \cdot I_{t-T,t,i} \times \prod_{\text{SL}}^{t} \frac{1}{1 + \text{Dis}_{i,t,s,r}} \right] \tag{15}$$

$C_{i,t,s,r}$ is the annual social cost of country $i$ in year $t$ to achieve the mitigation scenario from the SSP scenario $s$ and RCP scenario $r$. $L_{t,r,\text{mitigation}}$ is the loss of global GDP in year $t$ in climate change mitigation. The time series of $L_{t,r,\text{mitigation}}$ is derived by linear interpolation from the data points in 2020, 2030, 2050, and 2100. $R_{k,r,\text{mitigation}}$ is the ratio of the regional cost in region $k$ relative to the global cost of climate change mitigation (Supplementary Table 2). $k$ represents five regions: OECD 1990, Asia, Middle East and Africa, Latin America, and Economics in Transition. $\text{CC}_{i,T,s,r}$ is the discounted lifetime cost of the age cohort born in year $T$ from country $i$.

**Calculation net gain of GDP per capita during the lifetime.** We calculated the net gain of GDP per capita for an age cohort from climate change mitigation from the following equations:

$$\text{CGDP}_{i,T,s,r} = \sum_{t=\text{SL}}^{\text{EL}} \left[ \text{GDP}_{i,t,s,r} \cdot I_{t-T,t,i} \times \prod_{\text{SL}}^{t} \frac{1}{1 + \text{Dis}_{i,t,s,r}} \right] \tag{16}$$

$$\text{Net}_{i,T,s,r} = \text{CC}_{i,Ts,r} - \text{CB}_{i,T,s,r} \tag{17}$$

$$\text{RGDP}_{i,T,s,r} = \frac{\text{Net}_{i,T,s,r}}{\text{CGDP}_{i,T,s,r}} \tag{18}$$

$\text{CGDP}_{i,T,s,r}$ is the cumulative GDP per capita for the age cohort born in year $T$ in country $i$ under the baseline scenario of SSP $s$ and RCP $r$. $\text{Net}_{i,T,s,r}$ is the net gain of GDP per capita for the age cohort born in year $T$ in country $i$ by achieving the mitigation scenario from the SSP $s$ and RCP $r$. $\text{RGDP}_{i,T,s,r}$ is the percentage change of GDP per capita for the age cohort born in year $T$ in country $i$ in the climate change mitigation.

Calculation of breakeven year

$$\text{BY}_{\text{SY},i} = \begin{cases} T_{\text{SY}}, \text{ if } \begin{cases} \text{Net}_{i,T_{\text{SY}},s,r} > 0 \\ \text{Net}_{i,T_{\text{SY}}-1,s,r} < 0 \end{cases} \\ \text{nonexistence, Net}_{i,T,s,r} < 0 \text{ and } T \in [\text{SL, EL}] \end{cases} \tag{19}$$

$\text{BY}_{\text{SY},i}$ is the age cohort in country $i$ that breaks even the lifetime cost and benefit of the studied age cohorts (0–100 years old) in the year of SY (2020–2100). $T_{\text{SY}}$ is the age of the cohort in the year of SY. If none of the studied age cohort breaks even the lifetime cost and benefit, the age of the breakeven generation is defined as nonexistence.

**Uncertainty test.** The uncertainty of our calculation originates from the use of SSP and RCP scenarios, discount rates, the parameters in the equations, and the model specification of Eq. (6).

In this study, the RCP6.0 and SSP4 scenarios are used as the business as usual scenario, which is consistent with the global temperature increase and economic development under current policy by recent studies[32]. To analyze the uncertainty of different RCP and SSP scenarios, the social benefits under RCP4.5, RCP6.0 and RCP8.5 and all five SSP scenarios were calculated.

For the fixed discount rates, we consider 3% and 5% scenarios. For the growth-adjusted discount rate, we assumed that $\rho \in \{2, 1\}$ and $\mu \epsilon \{2\}$. All possible combinations of $\rho$ and $\mu$ are considered to test the sensitivity of discount rates on our results.

The uncertainty of $\beta_1, \beta_2$ in Eq. (6) is analyzed by bootstrapping using 1000 sets of parameter values. We assume that $L_{t,r,\text{mitigation}}$ and $r_{k,r,\text{mitigation}}$ in Eq. (13) follow the triangle distribution. We did 1000 times of simulation to test the uncertainty caused by parameters.

The results of the uncertainty test are provided online (https://climate-change.shinyapps.io/generation_disparity/).

The influence of temperature on GDP growth includes the contemporary and long-term effects. Rather than using pooled data, the model specification can also differentiate rich and poor countries. In Supplementary Fig. 2, we show the uncertainty caused by using different function forms of temperature and GDP growth.

**Reporting summary.** Further information on research design is available in the Nature Research Reporting Summary linked to this article.

## Data availability

The income distribution data used in this study are available in the OECD database (https://www.oecd.org/social/income-distribution-database). The life expectancy and age structure data are available in World Bank Open Data (https://data.worldbank.org/) and World Population Prospect (https://population.un.org/wpp/DataQuery/). The data used for replicating our analysis have been deposited in the Data for paper Economic disparity among generations under Paris Agreement[37] database under accession code https://doi.org/10.5281/zenodo.5103739. The data used to generate the interactive website have been deposited in the Data and code for the shiny app of the paper Economic disparity

among generations under Paris Agreement[38] database under the accession code https://doi.org/10.5281/zenodo.5104877. The data used to generate Figs. 1–3 are provided in the Supplementary Dataset. Data visualization for the analysis of other SSP and RCP scenarios can be found on our shinny app (https://climate-change.shinyapps.io/generation_disparity/). Source data are provided with this paper.

## Code availability

R 3.6.1 and MATLAB 2019b are used to process the data. R 3.6.1 and Origin 2019 are used for data visualization. All the scripts used in our data collection, data analysis, and data visualization are available at https://github.com/climate-change-ucsb/generation-disparity.

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

## Acknowledgements

We appreciate the helpful comments from Kyle Meng and Antony Millner. We thank the advice from Qian Gao, Yang Qiu, and Jiajia Zheng.

## Author contributions

H.Y. performed the research and analyzed the data. S.S. conceived the idea and supervised the work. H.Y. and S.S. designed the study and wrote the manuscript.

## Competing interests

The authors declare no competing interests.
