## [Peer Review File · Nature Communications]

Reviewer comments, first round –

Reviewer #1 (Remarks to the Author):

paris-report.pdf – [see next page]

Reviewer #2 (Remarks to the Author):

This paper looks at a fascinating question: what are the generational differences in costs and benefits of action on climate change? Much of the political momentum for action comes from younger generations, notably from wealthier countries, but are these groups acting in self interest? The answers will raise (and this paper does raise) interesting implications for whether cost-benefit frameworks actually reveal insight into political action. They also raise intriguing questions about whether, if people knew more about costs and benefits, they would be mobilizing politically in the ways they are today. The generational calculations are new as far as i can tell and a useful complement to political calculations that tend to use only the country and the region as a unit of analysis

A big caveat: it is important that this be reviewed by someone who knows the Burke et al method and also by someone working with the IAM techniques. I am neither.

I found the discussion compelling for the questions it raises and am sure there will be many disagreements, which is fine.

i have two concerns that I'd like to see addressed before publication.

First, most of the "work" in the paper seems to be done by the damage function. i saw no attention to possible variations in the allocation of costs across (or within) countries. Some of this is not tractable (e.g., within country cross-generational allocation of costs) but the ways the Paris process will lead to bargaining in allocation of costs seems pretty important.

Second, there is little attention to how uncertainty in the damage function could affect results.

beyond those two points, i found the discussion very thin--what might these kinds of results do for countries/governments as they try to make action on climate change more sustainable politically? will this kind of information make action harder to mobilize--because groups traditionally pressuring for action will care less? these kinds of questions are latent in this kind of analysis yet not engaged at all.

If you wish to transfer your manuscript to Scientific Reports, please use our <https://mts-ncomms.nature.com/cgi-bin/main.plex?el=A5S3CBON6A4GBky4X6A9ftdbSZe2VcEYtja3ZoIBckX1gZ> manuscript transfer portal to initiate the transfer to this journal (or to another journal of your choice in the Nature Research portfolio). If you transfer to Nature-branded journals or to the Communications journals, you will not have to re-supply manuscript metadata and files. This link can only be used once and remains active until used.

All Nature Research journals are editorially independent, and the decision to consider your manuscript will be taken by their own editorial staff. For more information, please see our http://www.nature.com/authors/author_resources/transfer_manuscripts.html?WT.mc_id=E

MI_NPG_1511_AUTHORTRANSF&WT.ec_id=AUTHOR">manuscript transfer FAQ page.
Note that any decision to opt in to In Review at the original journal is not sent to the receiving journal on transfer. You can opt in to *In Review at receiving journals that support this service by choosing to modify your manuscript on transfer. In Review is available for primary research manuscript types only.*

Referee report for Nature: Communications

Economic disparity among generations under Paris Agreement

- 1) My primary comment is that the quantitative aspect of the exercise needs to be improved, but I am unsure if this is possible to do in a satisfactory way. Below are a few comments and questions I have on this.
 - a. When are these costs expected to be incurred? I understand you are using the IPCC numbers but a plot of time series in the supplement would be helpful. Same thing for the benefits.
 - b. Even if the costs are nominally incurred at some year X , governments can borrow to finance and shift costs to the future. Similarly, we can finance expenditures to offset the impacts of climate change and bear the true cost later. This is what I think is the trickiest part of this exercise.
 - i. Even if we do not account for borrowing, the cost burden depends on how we raise money to pay for mitigation. Is it income taxes? Consumption taxes? Reductions in benefits? All of these have very different intergenerational impacts.
 - ii. These costs in the IPCC report are generally coming from representative agent IAMs with no attention to intergenerational issues (i.e. there's multiple/infinite possible paths to hit the Paris agreement depending on how welfare is optimized in these models). It is not really a good fit to use these estimates to study intergenerational problems.
 - c. Assigning GDP evenly across age cohorts can be improved upon. If we think of GDP in terms of production or as a measure of spending, it's clearly not evenly distributed.
 - d. Eq (13) how do you choose the regional ratio cost?
 - e. A bit more description on how the costs are computed in IPCC AR5 would be nice (which IAMs? What are the welfare criteria in these IAMs?).

- 2) There's a pretty extensive literature looking at this intergenerational issue that should be referenced in the paper.
- Andersen, Torben M., Joydeep Bhattacharya, and Pan Liu. "Resolving intergenerational conflict over the environment under the Pareto criterion." *Journal of Environmental Economics and Management* 100 (2020): 102290.
 - Karp, Larry, and Armon Rezai. "The political economy of environmental policy with overlapping generations." *International Economic Review* 55, no. 3 (2014): 711-733.
- 3) The paper is missing a lot of the intuition for what is driving everything. I suspect everything is because wealthier countries tend to be in colder regions which are neutral or possibly even benefit from climate change under the BHM response function. This explains why richer countries have positive or non-existent generation gaps (incurring costs for little or no benefit for any generation).
- 4) The last paragraph calling for transferring of benefits seems odd and tacked on. There's a lot of specific and reasonable ways to ameliorate intergenerational and cross-country inequality under the Paris agreement like just transferring costs across countries today by changing who bears the burden of abatement.
- 5) Another interesting angle is to consider how older generations have already extracted benefits by emitting CO₂ and thus bearing an outsized burden now may actually have positive effects on lifetime equity across generations.

Responses to the reviewer comments

Color code:

Comments received are marked in black.

Our responses are marked in blue.

Green texts are the original text in the previous version of the manuscript.

Red texts are new additions.

Referee 1

1) My primary comment is that the quantitative aspect of the exercise needs to be improved, but I am unsure if this is possible to do in a satisfactory way. Below are a few comments and questions I have on this.

a. When are these costs expected to be incurred? I understand you are using the IPCC numbers but a plot of time series in the supplement would be helpful. Same thing for the benefits.

Thank you for the comment. As we noted in our manuscript in Lines 50 - 51, the costs and benefits are expected to occur from 2020-2100. In response to your comment, we have created a new figure and included it as Figure 1 in Extended Data.

The costs and benefits of climate change mitigation are occurred within 2020 – 2100 (Extended Data Fig 1).

[Lines 50-51]

b. Even if the costs are nominally incurred at some year X, governments can borrow to finance and shift costs to the future. Similarly, we can finance expenditures to offset the impacts of climate change and bear the true cost later. This is what I think is the trickiest part of this exercise.

We agree that shifting costs to the future and financing expenditures can reduce the damage from climate change mitigation.

However, what we are trying to understand is the balance between the costs and benefits of climate change mitigation for each birth cohort in the *absence* of burden-shifting policy across generations under the *current IPCC policy scenarios*. What we are doing here is a scenario analysis for these current IPCC policy scenarios, which do not involve the financial or fiscal policies to solve intergenerational issue. Although these scenarios are not likely to be the future policy, they are relevant to the policy design today. Understanding the intergenerational disparity under these current policy scenarios is critical to provide the insights to future policies.

In summary, the scenario analysis is a stress test for these IPCC policy scenarios. These scenarios are not predictions, but their consequences are still useful to gain insights on the different economic incentives by birth cohorts in the absence of intergenerational burden-shifting policies. Here, we are examining the implications and political feasibility of current policy proposals instead of simulating the future policy trajectory in the future.

In our revised manuscript, we clarified that we are doing a scenario analysis of the current policy proposals in Lines 27-29 and Lines 48-50.

The answers will raise intriguing questions on how people would be mobilizing towards climate change mitigation if they know more about their costs and benefits under the current policy scenarios.

[Lines 27-29]

Here, we aim to measure the cost-benefit disparity among generations in the climate change mitigation under the current Paris Agreement scenario, which does not consider the policies to address the intergenerational disparity.

[Lines 48-50]

i. Even if we do not account for borrowing, the cost burden depends on how we raise money to pay for mitigation. Is it income taxes? Consumption taxes? Reductions in benefits? All of these have very different intergenerational impacts.

We agree that how we raise money to pay for mitigation will have different intergenerational impacts.

However, firstly, in IAMs which are used to measure the costs of climate change under the current IPCC policy proposals, the money to pay for climate change mitigation is raised by

levying carbon tax on upstream CO₂ emitters. The income tax, consumption tax or reductions in benefits are not considered as the funding source for climate change mitigation in IAMs following the IPCC policy scenarios.

Secondly, the current IPCC policy scenarios gives no consideration to intergenerational disparity. Therefore, the taxation in IPCC policy scenarios is unrelated to generations.

Lastly, to measure the intergenerational distribution of the costs, in our revision, we improved our model by using the income distribution across generations to allocate the costs. In IAMs, the average GDP loss per capita is calculated to measure the costs of climate change mitigation. Using the OECD database, we calculated the relative ratio of the income by age cohort and the average income of the total population. We assumed that GDP loss per capita across generations is proportional to this relative ratio of the income. For further information about the methods, please see the response to *question c* and Lines 343 – 354 and Equations (11) and (15).

ii. These costs in the IPCC report are generally coming from representative agent IAMs with no attention to intergenerational issues (i.e. there's multiple/infinite possible paths to hit the Paris agreement depending on how welfare is optimized in these models). It is not really a good fit to use these estimates to study intergenerational problems.

We agree that the current IAM models assume that there are no policy responses to resolve intergenerational imbalances in the costs and benefits of climate change mitigation. However, firstly, what we are trying to understand through this research is the magnitude of cost-benefit imbalances across birth-cohorts and geographies in the *absence* of the policies to address them under the current policy scenarios. Current IAMs all use the IPCC scenarios, which does not incorporate the policy to address the intergenerational issue. Therefore, we do believe that not only the use of IAMs serves the purpose of our research but also that the absence of intergenerational optimization in IAMs is a necessary feature that enables our estimates. Again, our study is not meant to be the prediction of the future. We are merely asking the question, what would be the intergenerational and geographical distributions of climate mitigation costs and benefits, should the current IPCC trajectories and policy measures, which take no consideration on intergenerational imbalances, are to be followed.

Secondly, in these IAMs, general equilibriums or partial equilibrium models are generally used, which rely on exogenous socioeconomic inputs from the IPCC scenarios. Therefore, although IAMs have different optimization method, they follow the same IPCC scenario and thus makes their results comparable.

Lastly, to measure the intergenerational distribution of the costs, we allocated the GDP loss across age cohorts by using the income distribution across generations in our revision. In IAMs, the total GDP loss is calculated to measure the costs of climate change mitigation, and allows us to calculate the average GDP loss per capita of the total population. Using the OECD database, we calculated the relative ratio of the income by age cohort over the average income of the total population. We assumed that GDP loss per capita across generations is proportional to this relative ratio of the income. For further information about the methods, please see the response to *question c* and Lines 343 – 354 and Equations (11) and (15).

c. Assigning GDP evenly across age cohorts can be improved upon. If we think of GDP in terms of production or as a measure of spending, it's clearly not evenly distributed.

We agree with the reviewer that our assumption of uniform income distribution across age cohorts within the same timeframe is a weakness of our study. In order to address this problem, we have revised our model by introducing non-uniform income distribution across age cohorts using OECD statistics (<https://www.oecd.org/social/income-distribution-database.htm>). In the OECD database that we have used, the income distribution of some of the developing countries are not included. In that case we used the median income distribution of the developing countries reported in the OECD database as a proxy. We assume that the income distribution across age cohorts is fixed over time from 2020 – 2100. Please see the equation (3) in Lines 343 – 354, the Equation (11) in Line 405 and the Equation (15) in Line 434 in our manuscript and Table 1 in Supplementary information for details.

Income distribution

The income distribution across age cohort depends on the disposable income at each age group and the mean disposable income of the total population.

$$I_{t-T,t,i} = \frac{D_{t-T,i}}{\bar{D}_i} \quad (3)$$

where t is the future year. T is the birth year of the age cohort. $t-T$ is the age of an age cohort in year t . At year t , $D_{t-T,i}$ is the income of an age cohort born in year T in country i . \bar{D}_i is the average income in country i . At year t , $I_{t-T,t,i}$ is the ratio of the income for an age cohort born in year T versus the average income of the total population in country i . The data for the income distribution across age cohorts are collected from OECD database³⁰. The income distribution of some developing countries are not included in the OECD database. Their income distribution is assumed to be the median of the income in developing countries that are included in the OECD database. We assume the income distribution is consistent over time.

[Lines 343-354]

$$CB_{i,T,s,r} = \sum_{t=SL}^{EL} \left[B_{i,t,s,r} \cdot I_{t-T,t,i} \times \prod_{t=SL}^{EL} \frac{1}{Dis_{i,t,s,r}} \right] \quad (11)$$

$B_{i,t,s,r}$ is the annual social benefit of country i in year t under SSP scenario s and RCP scenario r . $CB_{i,T,s,r}$ is the discounted lifetime benefits of the age cohort born in year T from country i by achieving the mitigation scenario from the SSP scenario s and RCP scenario r . $Dis_{i,t,s}$ is the discount rate in country i in year t .

[Equation 11 in Line 405]

$$CC_{i,T,s,r} = \sum_{t=SL}^{EL} \left[C_{i,t,s,r} \cdot I_{t-T,t,i} \times \prod_{t=SL}^{EL} \frac{1}{Dis_{i,t,s,r}} \right] \quad (15)$$

$C_{i,t,s,r}$ is the annual social cost of country i in year t to achieve the mitigation scenario from the SSP scenario s and RCP scenario r . $CC_{i,T,s,r}$ is the discounted lifetime cost of the age cohort born in year T from country i .

[Equation 11 in Line 434]

d. Eq (13) how do you choose the regional ratio cost?

The regional ratio is collected from the IPCC report, which is the ratio of the regional GDP percentage loss versus the global GDP percentage loss. In the report, the world is divided into 5 regions: OECD, Asia, Middle East and Africa, Latin America, and Economies in Transition. In our original manuscript, we have introduced how we use the regional ratio in Lines 429 – 441.

For the data source, please see Page 457 in the IPCC report *Clarke L., et al., 2014: Assessing Transformation Pathways. In: Climate Change 2014: Mitigation of Climate Change. Contribution of Working Group III to the Fifth Assessment Report of the Intergovernmental Panel on Climate Change*. The link to the report is listed below:

https://www.ipcc.ch/site/assets/uploads/2018/02/ipcc_wg3_ar5_chapter6.pdf.

In our revision, we have added Extended Data Table 1 to show the regional ratio cost.

Extended Data Table 1. Regional relative cost of climate change mitigation versus global relative cost¹⁴.

The relative cost is computed as the cumulative costs of mitigation over the period 2020 – 2100 divided by cumulative GDP over that period.

Region	Median	25% percentile	75% percentile
OECD 1990	0.47	0.41	0.61
Asia	1.47	1.19	1.64
Middle East and Africa	2.29	1.79	2.79
Latin America	0.99	0.92	1.15
Economies in Transition	1.99	1.39	2.49

Calculating the cost of climate change mitigation

We used the loss of GDP from the 2014 IPCC report to calculate the cost of climate change mitigation.

$$C_{i,t,s,r} = G_{i,t,s,r} \cdot L_{i,t,r,mitigation} \quad (13)$$

$$L_{i,t,r,mitigation} = L_{t,r,mitigation} \cdot R_{k,r,mitigation}, i \in k \quad (14)$$

$$CC_{i,T,s,r} = \sum_{t=SL}^{EL} \left[C_{i,t,s,r} \cdot I_{t-T,t,i} \times \prod_{t=SL}^{EL} \frac{1}{Dis_{i,t,s,r}} \right] \quad (15)$$

$C_{i,t,s,r}$ is the annual social cost of country i in year t to achieve the mitigation scenario from the SSP scenario s and RCP scenario r . $L_{t,r}$ is the loss of global GDP in year t in climate change mitigation. The time series of $L_{t,r}$ is derived by liner interpolation from the data points in 2020, 2030, 2050 and 2100. $R_{k,r}$ is the ratio of the regional cost in region k relative to the global cost of climate change mitigation. k represent 5 regions: OECD 1990, Asia, Middle East and Africa, Latin America and Economics in Transition. $CC_{i,T,s,r}$ is the discounted lifetime cost of the age cohort born in year T from country i .

[Lines 429 – 441]

e. A bit more description on how the costs are computed in IPCC AR5 would be nice (which IAMs? What are the welfare criteria in these IAMs?).

We agree that more description about IAM would be useful. In our revision, we added an introduction to the IAMs used in IPCC AR5 in the main text and Supplementary Information. Please see Lines 32-40 of the main text and the Supplementary Information.

Main text

In this study, the cost of climate change mitigation refers to the gross domestic product (GDP) loss compared to the counterfactual scenario without climate change mitigation¹⁴.

To measure the loss of GDP, Integrated Assessment models (IAMs) are developed by

many research groups to couple energy, economy and climate. In these IAMs, the economic modules generally follow the general or partial equilibrium models¹⁵. Here, the data for the cost of climate change mitigation is summarized from several IAMs (https://www.iamcdocumentation.eu/index.php/IAMC_wiki) in the 2014 IPCC report. According to the report, the abatement cost of climate change mitigation range 2-6% of global GDP by 2100 relative to pre-Paris Agreement policy¹⁴.

[Lines 32-40]

Supplementary Information

1. Economic modules in IAMs

Here we classify the current IAMs into three categories based on the literature¹.

1.1 General Equilibrium model

Computational General Equilibrium (CGE) models are an algebraic representation of the intricate functioning of a market economy based on the economic equilibrium theory. By maximizing consumer utility and producer profits, the supply and demand are in equilibrium. The general way that CGE models are used for policy analysis is to change one or more of the exogenous parameters of the economy and compute the new equilibrium. Comparing the new counterfactual equilibrium to the initial equilibrium, including activity levels, prices and utility, provides insights about the effect of a “shock” on the economy.

IAMs using the CGE principles include: AIM/CGE, GEM, REMIND, WITCH and IMACLIM.

1.2 Partial Equilibrium model

Partial equilibrium analysis differs from general equilibrium modelling primarily by focusing on a specific market or sector. Partial equilibrium analysis is used extensively to estimate the impacts of climate change in different sectors of the economy.

IAMs using the partial equilibrium model include: GCAM, MESSAGE, TIAM-UCL.

1.3 Energy system models

Energy system models can broadly be classified as optimization models or simulation models. Optimization models use information on costs and constraints of technology characteristics to identify the “best”, “least-cost” or “optimal” technology. The consumer is assumed to be rational, and energy supplies are allocated to energy demands, based on minimum lifecycle technology costs.

IAMs using the energy system models include: MESSAGE, DNE21+.

2) There’s a pretty extensive literature looking at this intergenerational issue that should be referenced in the paper.

- Andersen, Torben M., Joydeep Bhattacharya, and Pan Liu. "Resolving intergenerational conflict over the environment under the Pareto criterion." *Journal of Environmental Economics and Management* 100 (2020): 102290.
- Karp, Larry, and Armon Rezai. "The political economy of environmental policy with overlapping generations." *International Economic Review* 55, no. 3 (2014): 711-733.

We agree that these are relevant literature for our paper. In our revision, we have added them in our paper. Please see Lines 22-24 and 243-245.

3) The paper is missing a lot of the intuition for what is driving everything. I suspect everything is because wealthier countries tend to be in colder regions which are neutral or possibly even benefit from climate change under the BHM response function. This explains why richer countries have positive or non-existent generation gaps (incurring costs for little or no benefit for any generation).

We agree with the reviewer’s comments. The BHM response function used the quadratic function form. Therefore, in colder regions, the temperature increase has neutral or even positive effects on the economy. That colder regions may be neutral or even benefit from climate change has been supported in many research, for example:

1. Nordhaus, William D., and Joseph Boyer. Warming the world: economic models of global warming. MIT press, 2000.
2. Dell, Melissa, Benjamin F. Jones, and Benjamin A. Olken. Climate change and economic growth: Evidence from the last half century. No. w14132. National Bureau of Economic Research, 2008.
3. Tol, Richard SJ, et al. "Distributional aspects of climate change impacts." Global Environmental Change 14.3 (2004): 259-272.
4. Ricke, Katharine, et al. "Country-level social cost of carbon." Nature Climate Change 8.10 (2018): 895-900.
5. Diffenbaugh, Noah S., and Marshall Burke. "Global warming has increased global economic inequality." Proceedings of the National Academy of Sciences 116.20 (2019): 9808-9813.
6. Tol, Richard SJ. "A social cost of carbon for (almost) every country." Energy Economics 83 (2019): 555-566.

Following the reviewer's comment, we have added additional description on the drivers behind the observation that we are making and cited relevant literature accordingly. Please see Lines 129-130.

4) The last paragraph calling for transferring of benefits seems odd and tacked on. There's a lot of specific and reasonable ways to ameliorate intergenerational and cross-country inequality under the Paris agreement like just transferring costs across countries today by changing who bears the burden of abatement.

We agree with the reviewer's comments. We have added a discussion in the Discussion section, where we added the implication of our results about the tax on carbon and renewable assets. For the carbon tax, our results indicate that the consumption tax and income tax are more preferable to tax on upstream producers. This is because tax on upstream producers does not consider the heterogeneity of generations. Please see Lines 237 – 246.

Our results provide a new insight on intergenerational equity of climate change policy. Closing the economic disparity among age cohorts, different climate policies may be applied to different age cohorts. Carbon tax, for example, is one of the major policy levers used to stimulate climate change mitigation. However, the carbon tax is commonly applied as a production tax, which gives no consideration to the cost-benefit disparity

across age cohorts. But aligning carbon tax to income or consumption may more equitably distribute the cost of climate change mitigation across age cohorts. It is also shown that the increase in renewable asset price can alleviate the intergenerational disparity in climate change mitigation, given that different age cohorts hold varying amounts of renewable assets²⁸. Therefore, the tax on renewable assets can also be adjusted according to the intergenerational disparity.

[Lines 237 – 246]

5) Another interesting angle is to consider how older generations have already extracted benefits by emitting CO₂ and thus bearing an outsized burden now may actually have positive effects on lifetime equity across generations.

We agree that the older generations have already extracted benefits by emitting GHG emissions, which have positive effects on lifetime income. We also understand that historical emissions and their responsibilities across age cohorts is a complex topic outside of the scope for this paper. What we are trying to understand through this research is the magnitude of the *future* cost-benefit imbalances due to climate change mitigation across birth-cohorts in the absence of the policies to address them, following the IPCC trajectory. What we are most interested in is how the cost-benefit imbalance across age cohorts could influence their behaviors in future climate change mitigation. Measuring the historical benefits of older generations by emitting CO₂ is an important issue, but it is out of the scope for our study.

Referee 2

1. First, most of the "work" in the paper seems to be done by the damage function. I saw no attention to possible variations in the allocation of costs across (or within) countries. Some of this is not tractable (e.g., within-country cross-generational allocation of costs) but the ways the Paris process will lead to bargaining in allocation of costs seems pretty important.

Thank you for the suggestion.

In our original manuscript, we have introduced how we considered the regional variations of the costs in Lines 429-441. As no IAMs have estimated the cost at country level, we can only consider the regional variation of the cost. Based on the IPCC report (https://www.ipcc.ch/site/assets/uploads/2018/02/ipcc_wg3_ar5_chapter6.pdf, Page 457), there are 5 regions in the IAMs used: Asia, OECD countries, Middle East and Africa, Latin America, and Economics in Transition.

In our revision, we added Extended Data Table 1 to show the regional costs. We also introduced the income distribution across age cohorts to allocate the benefits and costs across generations within countries. Please see Lines 343-354 and equation (15) for further information.

Income distribution

The income distribution across age cohort depends on the disposable income at each age group and the mean disposable income of the total population.

$$I_{A,t,i} = \frac{D_{t-SY+A,i}}{\bar{D}_i} \quad (3)$$

where t is the future year which is larger than the studied year SY . A is the age of the age cohort in the studied year SY . Therefore, $t-SY+A$ is the age of an age cohort in year t . At year t , $D_{t-SY+A,i}$ is the income of an age cohort born in year SL in country i . \bar{D}_i is the average income in country i . At year t , $I_{a,t,i}$ is the ratio of the income for an age cohort born in SL versus the average income in country i . The data for the income distribution across age cohorts are collected from OECD database²⁸. The income distribution of some

developing countries are not included in the OECD database. Their income distribution is assumed to be the median of the income in developing countries that are included in the OECD database. We assume the income distribution is consistent over time.

[Lines 343-354]

We used the loss of GDP from the 2014 IPCC report to calculate the cost of climate change mitigation.

$$C_{i,t,s,r} = G_{i,t,s,r} \cdot L_{i,t,r,mitigation} \quad (13)$$

$$L_{i,t,r,mitigation} = L_{t,r,mitigation} \cdot R_{k,r,mitigation}, i \in k \quad (14)$$

$$CC_{i,T,s,r} = \sum_{t=SL}^{EL} \left[C_{i,t,s,r} \cdot I_{t-T,t,i} \times \prod_{t=SL}^{EL} \frac{1}{Dis_{i,t,s,r}} \right] \quad (15)$$

$C_{i,t,s,r}$ is the annual social cost of country i in year t to achieve the mitigation scenario from the SSP scenario s and RCP scenario r . $L_{t,r}$ is the loss of global GDP in year t in climate change mitigation. The time series of $L_{t,r}$ is derived by linear interpolation from the data points in 2020, 2030, 2050 and 2100. $R_{k,r}$ is the ratio of the regional cost in region k relative to the global cost of climate change mitigation. k represent 5 regions: OECD 1990, Asia, Middle East and Africa, Latin America and Economics in Transition. $CC_{i,T,s,r}$ is the discounted lifetime cost of the age cohort born in year T from country i .

[Lines 429-441]

2. Second, there is little attention to how uncertainty in the damage function could affect results.

Response:

Although we have included some of the details on uncertainties in our original manuscript, we agree that the uncertainty aspect of our model can be further elaborated.

In the original manuscript, we mainly discussed the results based on the short-term damage function, which is more commonly used in the other research. We discussed the uncertainties in relation to the function form, scenarios, and parameters used in our model (Lines 463-482). We also analyzed the uncertainty on an interactive website https://climate-change.shinyapps.io/generation_disparity/.

In our revision, more information about the uncertainty of the damage function is added. The uncertainty of the damage function is shown in Extended Data Figure 2. We discussed how the uncertainty of function would affect the generational costs and benefits in Lines 81-99, the breakeven generation in Lines 124-133 and Lines 138-142, the intergenerational disparity in Lines 190-195. In the discussion, we discussed how the uncertainty could influence the mobilization of climate change mitigation in Lines 221-230.

In high- and upper-middle-income countries, the trend of lifetime GDP per capita by age cohort is sensitive to the model specification to measure the benefits of climate change mitigation. When using the short-term BHM damage function to measure the lifetime benefits (short-term benefits), which is commonly used by other research^{15,16,19,23,24}, the age cohorts in many high- and upper-middle-income countries still incur a net reduction with the progression of birth year, including the age cohorts born in 2020. The age cohorts in high and upper-middle-income countries, on average, barely gain any net benefits throughout the birth years considered. On average, cohorts in high-income countries lose 0-2% of lifetime GDP per capita, and those in upper-middle-income countries lose 0-3% of lifetime GDP per capita.

However, when using the long-term BHM damage function to measure the lifetime benefits (long-term benefits), the net gain of GDP per capita increase in high- and upper-middle-income countries with the progress of birth years. The age cohorts born in 1960 in high-income countries (Fig 1e), and the age cohorts born in 1980 in upper-middle-income countries (Fig 1f) start to show net gains of average lifetime GDP per capita in climate change mitigation.

When using the long-term BHM damage function to measure the benefits of climate change mitigation, the uncertain range is much wider than that using the short-term damage function (Extended Data Fig. 2). Due to the large uncertainties of the long-term damage

function and lack of robust evidence for the long-term benefits, the short-term benefits of climate change mitigation are more commonly discussed in the current literature^{24,18,23}.

[Lines 81-99]

In high-income countries, the breakeven generation are born prior to 1980 in Spain, Australia and Saudi Arabia. The birth years of the breakeven generation in Europe, Canada and the United States are sensitive to the model specification. Using the short-term lifetime benefits, none of the studied age cohorts (age cohorts born between 1920-2020) become the breakeven generation in Canada and most Western European countries, and the breakeven generation in the United States is born in 1994. This is because, in colder regions, the temperature increase has neutral or even positive effects on the economy in the short-term^{23,25-27}. When considering the long-term lifetime benefits, the breakeven generation are born prior to 1970 in Canada, the United States and Western Europe, and more than three quarters of the population are born before the breakeven generation.

[Lines 124-133]

The birth years of the breakeven generation in Asia are uncertain due to the model specification, ranging from 1970 to nonexistence. In China, none of the age cohorts break even the costs and benefits considering the short-term benefits, but two thirds of the population are born after the breakeven generation considering the long-term benefits.

[Lines 138-142]

Most of our results are robust with the model specification (Extended Data Fig 5). Using the long-term benefits, we still find that the intergenerational disparity index is increasing over time, and becomes larger in countries with lower income. In low-income countries, the median of the intergenerational disparity index increases from 0.04 (2020) to 0.34 (2100). In lower-middle-income countries, upper-middle-income countries and low-income countries, the median intergenerational disparity index increases from 0.05 to 0.31, 0.28 and 0.23 during 2020 – 2100.

[Lines 190-195]

Our results based on the short-term damage function of climate change, which has been used as the default model in many previous studies^{24,19,23,15}, show that no age cohorts enjoy net benefit from climate change mitigation in most Western European countries and China in 2020. If we follow the assumption that the level of support toward climate change mitigation is positively correlated to the net lifetime benefits, our results using the short-term damage function indicate that younger age cohorts in Western Europe could change their current attitudes and actions towards climate change mitigation if they are aware of their future cost-benefit status. However, using the long-term damage function, we find that younger age cohorts in Western Europe, China and USA also benefit from climate change mitigation, indicating their sustainable support to climate change mitigation.

[Lines 221-230]

3. Beyond those two points, I found the discussion very thin--what might these kinds of results do for countries/governments as they try to make action on climate change more sustainable politically? Will this kind of information make action harder to mobilize--because groups traditionally pressuring for action will care less? These kinds of questions are latent in this kind of analysis yet not engaged at all.

We agree that such discussion points were not fully exploited in our original manuscript. In our revision, we first discussed how the intergenerational disparity could affect the action of climate change mitigation in Lines 211-230. We also discussed the implication of our results on carbon tax and tax on assets. For the carbon tax, our results indicate that the consumption tax and income tax are more preferable to production tax. This is because production tax does not consider the heterogeneity of income across generations. As the increasing of asset price can help age cohorts who are not benefiting from climate change mitigation gain additional benefits, the tax on assets can also consider the generational disparity (Lines 237-246).

If one subscribes to the idea that the level of support to climate change mitigation is positively correlated to the life-time benefits from climate change mitigation, lower-

income (lower-middle- and low-income) countries are likely to see more support to climate change mitigation from older generations, because more than half of the population are born before the breakeven generation in most of low-income countries. In the same vein, the countries in Latin America, Western Asia and Australia are likely to see more support to climate change mitigation by older generations among higher-income (high- and upper-middle-income) countries. Under this assumption, the climate change mitigation effort is likely to face challenges in Eastern Europe, because, regardless of the model specification, less than half of the population in Eastern Europe are likely to gain net benefits from climate change mitigation.

Our results based on the short-term damage function of climate change, which has been used as the default model in many previous studies^{24,19,23,15}, show that no age cohorts enjoy net benefit from climate change mitigation in most Western European countries and China in 2020. If we follow the assumption that the level of support toward climate change mitigation is positively correlated to the net lifetime benefits, our results using the short-term damage function indicate that younger age cohorts in Western Europe could change their current attitudes and actions towards climate change mitigation if they are aware of their future cost-benefit status. However, using the long-term damage function, we find that younger age cohorts in Western Europe, China and USA also benefit from climate change mitigation, indicating their sustainable support to climate change mitigation.

[211-230]

Our results provide a new insight on intergenerational equity of climate change policy. Closing the economic disparity among age cohorts, different climate policies may be applied to different age cohorts. Carbon tax, for example, is one of the major policy levers used to stimulate climate change mitigation. However, the carbon tax is commonly applied as a production tax, which gives no consideration to the cost-benefit disparity across age cohorts. But aligning carbon tax to income or consumption may more equitably distribute the cost of climate change mitigation across age cohorts. It is also shown that the increase in renewable asset price can alleviate the intergenerational disparity in climate change mitigation, given that different age cohorts hold varying

amounts of renewable assets²⁸. Therefore, the tax on renewable assets can also be adjusted according to the intergenerational disparity.

[Lines 237-246]

Reviewer comments, second round –

Reviewer #2 (Remarks to the Author):

The revisions are good. Publishable as is.